# Identification of fresh leaves of Anji White Tea: S-YOLOv10-ASI algorithm fusing asymptotic feature pyra-mid network

Chunhua Yang[1,2], Wenxia Yuan[2], Qiang Zhao[1], Zejun Wang[2], Bowu Song[1], Xianqiu Dong[3], Yuandong Xiao[2], Shihao Zhang[1,2]*, Baijuan Wang ![ORCID][2]*

1 College of Mechanical and Electrical Engineering, Wuhan Donghu College, Wuhan, China, 2 College of Tea Science, Yunnan Agricultural University, Kunming, China, 3 Xuzhou Taxation Bureau of the State Administration of Taxation, Xuzhou, China

* zhangshihao@wdu.edu.cn (SZ); wangbaijuan2023@163.com (BW);

## Abstract

This study proposes the S-YOLOv10-ASI algorithm to improve the accuracy of tea identification and harvesting by robots, integrating a slice-assisted super-reasoning technique. The algorithm improves the partial structure of the YOLOv10 network through space-to-depth convolution. The Progressive Feature Pyramid Network minimizes information loss during multi-stage transmission, enhances the saliency of key layers, resolves conflicts between objects, and improves the fusion of non-adjacent layers. Intersection over Union (IoU) is used to optimize the loss function calculation. The slice-assisted super-reasoning algorithm is integrated to improve the recognition ability of YOLOv10 network for long-distance and small-target tea. The experimental results demonstrate that when compared to YOLOv10, S-YOLOv10-ASI shows significant improvements across various metrics. Specifically, Bounding Box Regression Loss decreases by over 30% in the training set, while Classification Loss and Bounding Box Regression Loss drop by more than 60% in the validation set. Additionally, Distribution Focal Loss reduces by approximately 10%. Furthermore, Precision, Recall, and mAP have all increased by 7.1%, 6.69%, and 6.78% respectively. Moreover, the AP values for single bud, one bud and one leaf, and one bud and two leaves have seen improvements of 6.10%, 7.99%, and 8.28% respectively. The improved model effectively addresses challenges such as long-distance detection, small targets, and low resolution. It also offers high precision and recall, laying the foundation for the development of an Anji White Tea picking robot.

## Instruction

Anji White Tea [1,2], renowned both in China and internationally, is celebrated for its distinctive flavor and exceptional quality. The region's unique natural conditions

**Data availability statement:** All relevant data for this study are publicly available from the IEEE DATAPort repository (http://doi.org/10.21227/eefx-a805).

**Funding:** This research was funded by the Major special projects of Yunnan Provincial Science and Tech-nology Department in 2023(202302AE090020), Key Projects of Basic Research Program of Yunnan Province in 2023 (202201AS070325) and Yunnan Provincial Science and Technology Department 2023 Rural Revitalization Science and Technology Special Project(202304BI090013). The funders had no role in study design, data collection and analysis, decision to publish, or preparation of the manuscript.

**Competing interests:** No authors have competing interests.

create an ideal environment for cultivation, but also present challenges for the mechanization and automation of tea harvesting. Traditional manual picking is not only labor-intensive and inefficient, but also easy to vary in quality due to manual operation. Moreover, recent years have seen an increasing problem with the aging of tea farmers and a shortage of labor, which has severely hampered the development of tea production. Therefore, applying mechanization and intelligent technologies to enhance the efficiency and quality of tea production has become crucial for improving tea picking processes.

With the advancement of agricultural science and technology, mechanized and intelligent systems have made significant strides, including developments such as tomato-picking robots [3,4] and intelligent irrigation systems. However, in tea production, especially in the tea picking process, the application of mechanization and intelligence is still in its infancy. Compared to robots for tomato picking, apple picking, and others [5,6], tea picking robots [7] face more stringent requirements for recognition and detection algorithms due to the dense growth of tea leaves, complex environments, and significant variations in lighting.

The advancement of deep learning technology [8,9] has significantly enhanced capabilities in target detection and image recognition tasks. YOLO (You Only Look Once) series network, as a representative of it, has become a research hotspot because of its efficient real-time processing ability and good detection accuracy. In 2023, our research team proposed the ShuffleNetv2-YOLOv5-Lite-E algorithm for Yunnan large-leaf tea tree picking [10]. The YOLOv5 network was deeply improved by network structure re-placement and channel pruning. A bud and two leaves recognition rate of 94.52% was achieved on edge devices. Lixue Zhu et al. [11] improved the YOLOv5 network through the ECANet (efficient channel attention network) module and the BiFPN (bidirectional feature pyramid network). The accuracy of tea bud detection reached 94.4%, and the recall rate reached 90.38%. Aiming at the problem of low detection accuracy caused by the complex background of tea, Shuang Xie et al. [12] by integrating deformable convolution, attention mechanism and improved spatial pyramid pooling, proposed an improved Tea-YOLOv8s algorithm. And the accuracy of tea bud detection reached 88.27%. Nevertheless, these methods fail to overcome real-world challenges including long-range detection, small targets, and low-resolution conditions.

To address these challenges, this study proposes an enhanced YOLOv10 architecture incorporating Space-to-Depth Convolution, which effectively preserves fine-grained details in long-distance and low-resolution imagery. The asymptotic feature pyramid network is used to optimize the YOLOv10 network framework, reduce the information loss of the network in multi-stage transmission, enhance the saliency of the key layer, alleviate the contradictory information of different objects, and optimize the fusion of non-adjacent layers. The Inner-IoU [13] optimization loss function calculation is used to improve the convergence speed of the model and enhance the universality of the model. The slice-assisted super-reasoning algorithm is integrated to improve the recognition ability of YOLOv10 network for long-distance and small-target tea. This study aims to develop a high-precision, robust visual recognition

model specifically optimized for Anji white tea harvesting robots, with particular emphasis on accuracy and generalization capability across varying field conditions. Thereby laying the groundwork for the creation of specialized robots for picking Anji white tea.

## Materials and methods

### Image collection and dataset preparation

The datasets for this study were collected on-site in Anji County, Huzhou City, Zhejiang Province. The acquisition device uses Canon EOS R5 camera with RF24–105mm lens to ensure high resolution and clarity of the original image. To improve the model's detection accuracy in real-world conditions, this study simulated the distance between the tea-picking robot and the tea tree. The shooting distance ranged from 20 to 100 centimeters, while the distance between the robot and the ground varied from 100 to 160 centimeters. The ISO is set to 200, the aperture is f/4.0, and the shutter speed is 1/300. In order to ensure that the model can accurately identify tea leaves under different illumination, different slopes and different angles, and has stronger universality in practical applications, a multi-period and multi-angle image acquisition method is adopted. The resolution of the image is 5152 pixels (horizontal) × 3864 pixels (vertical), and the storage format is jpg. In this study, the tea posture is divided into side angle and positive angle according to the different position of tea. The image acquisition device is located in the middle of the robot, and the height can be freely adjusted. When the tea leaves grow on both sides of the field of view, they are considered to be in a side view. The robot, equipped with an occlusal end effector, must adjust the effector's angle to approach the base of the tea leaves for harvesting. When the tea grows at the top of the field of view, the fresh leaves of the tea are considered to be a positive angle of view, and the occlusal end-effector can directly complete the picking, as shown in Fig 1.

To enhance the model's generalization capability in complex and low-light conditions [14,15] and better capture tea features from different distances and angles, as shown in Fig 2, the collected data was added to improve the performance and robustness of the final detection model under real-world operating conditions. To improve the model's recognition performance under diverse lighting conditions in the tea garden, the brightness of the images was randomly adjusted within a range of 0.5 to 1.5 times. Additionally, to enhance the model's performance in fuzzy conditions, the image contrast was randomly varied from 0.4 to 1.8 times. The random range is mainly set based on the actual environmental variations in the tea garden. Furthermore, to strengthen the model's ability to extract tea features at different distances and fields of view, the images were randomly scaled along both the x-axis and y-axis.

In this study, a total of 2265 original samples were collected, and 15855 images were enhanced. After screening, a total of 15000 images were selected. After annotating the image with Make Sense, it is randomly divided into training set, verification set and test set, with the number of 9000, 3000 and 3000 respectively. The data set is based on image brightness, image contrast and scaling, and the division results are shown in Table 1.

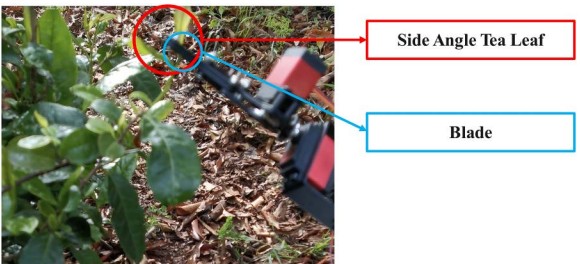

**Fig 1.  Side angle tea picking.**

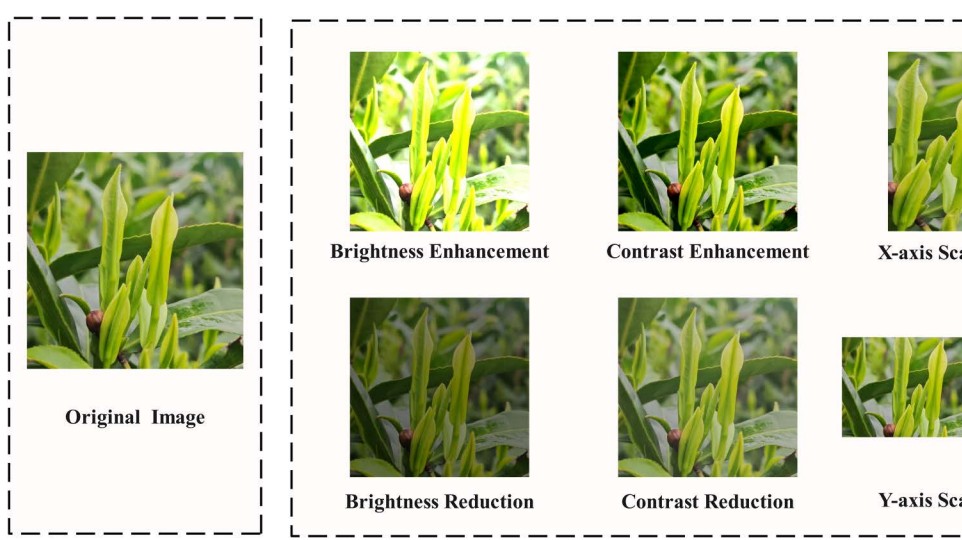

**Fig 2. Data augmentation.**

**Table 1. Table of dataset division.**

| Classification Conditions | Training Set | Validation Set | Test Set | Total |
|---|---|---|---|---|
| Original Image | 1264 | 513 | 479 | 2256 |
| Low Light | 1215 | 493 | 466 | 2174 |
| Strong Light | 1407 | 453 | 478 | 2338 |
| High Contrast | 1386 | 459 | 470 | 2315 |
| Low Contrast | 1415 | 434 | 348 | 2197 |
| X-axis Scaling | 1123 | 297 | 365 | 1785 |
| Y-axis Scaling | 1190 | 351 | 394 | 1935 |
| Total | 9000 | 3000 | 3000 | 15000 |

## YOLOv10 network improvements

YOLOv10 is the latest version of the YOLO series [16,17]. It is a deep learning network for target detection. The network demonstrates notable efficiency in real-time processing while maintaining robust detection accuracy. Nevertheless, the YOLOv10 network exhibits two notable limitations: (1) significant detail degradation when processing low-resolution images, (2) suboptimal recognition performance for small tea targets at long distances. To address these issues, this study employs Space-to-Depth Convolution to enhance specific components of the YOLOv10 network, reducing the loss of detail in long-distance and low-resolution images. The Asymptotic Feature Pyramid Network (AFPN) is used to optimize the YOLOv10 framework. It reduces information loss during multi-stage transmission, improves the saliency of key layers, alleviates conflicting information between objects, and enhances the fusion of non-adjacent layers. The Inner-IoU optimization loss function calculation is used to improve the convergence speed of the model and enhance the universality of the model. The slice-assisted super-reasoning algorithm is integrated to improve the recognition ability of YOLOv10 network for long-distance and small-target tea. The improved YOLOv10 network is shown in Fig 3. The enhancements to the Space-to-Depth Convolution primarily target the layers subsequent to the second through fifth convolutional layers within the backbone. Regarding the advancements in the Asymptotic Feature Pyramid Network, the primary focus lies on

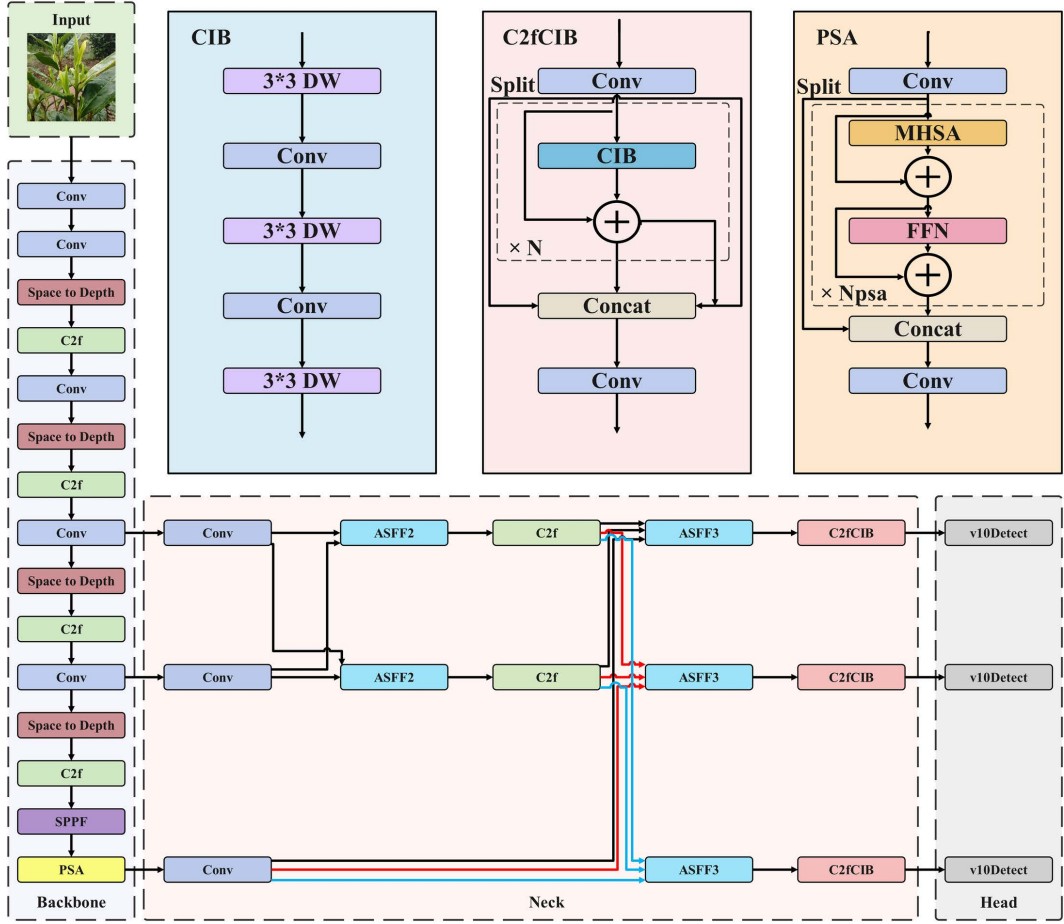

**Fig 3. Improved YOLOv10 network structure.**

the neck section. The Slicing Aided Hyper Inference Algorithm is positioned between the input and the backbone during the model's detection phase.

## Space-to-depth convolution enhancement

During deployment, tea-picking robots often capture blurred images [18–20] due to various factors, including high movement speed, mechanical vibrations, variable lighting conditions, and suboptimal focal length adjustments. The YOLOv10 network tends to lose significant detail when processing low-resolution images or distant tea leaves. To address these limitations, this study employs Space-to-Depth Convolution to enhance specific components of the YOLOv10 architecture. As shown in Fig 4, the Space-to-Depth Convolution is mainly composed of a SPD (Space-to-Depth) layer and a non-striding convolution layer. The SPD divides the input feature map into multiple sub-feature maps according to a certain stride, and stitches these sub-feature maps on the channel dimension, thereby reducing the space size but retaining all information. Non-strided convolution is applied to reduce the number of channels while retaining discriminative feature information.

As shown in Equations (1–3), where X represents the original feature map, and the size is $S \times S \times C_1$. $f_{x,y}$ represents the sub-feature map, which is generated by the proportional division of the original feature map, so each sub-feature map needs to down sample the original feature map. For example, when scale=2, four subgraphs $f_{0,0}$, $f_{1,0}$, and $f_{1,1}$ can be obtained, and

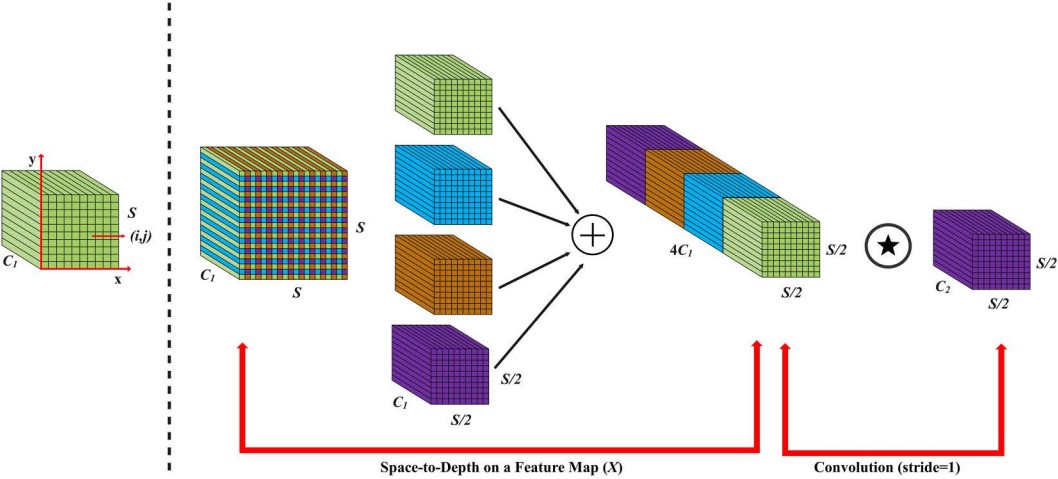

**Fig 4. Space-to-depth convolution.**

the size is $\frac{S}{2} \times \frac{S}{2} \times C_1$. As shown in Equations 4 and 5, $X'$ represents the sub-feature map after linking along the channel dimension. $X''$ represents the transformed $X'$, and a non-stepped convolutional layer with a $C_2$ filter is added during the conversion process. The non-stepped convolutional layer is mainly used to preserve all discriminant feature information.

$$
\begin{cases}
f_{0,0} = X[0 : S : scale,\ 0 : S : scale] \\
f_{1,0} = X[1 : S : scale,\ 0 : S : scale] \\
\qquad\qquad\qquad\qquad\qquad \ldots \\
f_{scale-1,0} = X[scale-1 : S : scale,\ 0 : S : scale]
\end{cases}
\tag{1}
$$

$$
\begin{cases}
f_{0,1} = X[0 : S : scale,\ 1 : S : scale] \\
\qquad\qquad\qquad\qquad\qquad \ldots \\
f_{scale-1,1} = X[scale-1 : S : scale,\ 1 : S : scale]
\end{cases}
\tag{2}
$$

$$
\begin{cases}
f_{0,scale-1} = X[0 : S : scale,\ scale-1 : S : scale] \\
\qquad\qquad\qquad\qquad\qquad \ldots \\
f_{scale-1,scale-1} = X[scale-1 : S : scale,\ scale-1 : S : scale]
\end{cases}
\tag{3}
$$

$$
X' = \left( \frac{S}{sacle}, \frac{S}{sacle}, sacle^2 C_1 \right)
\tag{4}
$$

$$
X'' = \left( \frac{S}{sacle}, \frac{S}{sacle}, C_2 \right)
\tag{5}
$$

## Asymptotic feature pyramid network improvement

During field operations, variations in the distance between the data acquisition device and tea leaves, along with inherent differences in leaf sizes, can lead to feature loss in the YOLOv10 network. This adversely affects the fusion of non-adjacent layers. To address these issues, this study introduces an enhanced Asymptotic Feature Pyramid Network (AFPN) to optimize the YOLOv10 architecture. This improvement enables effective fusion between non-adjacent layers by

progressively incorporating higher-level features. In the refined YOLOv10 network, the initial stage focuses on integrating low-level features, which are rich in detailed information.

The high-level feature fusion occurs in the final stage, as illustrated in Fig 5. In this schematic, blue arrows represent convolution operations, yellow arrows indicate adaptive spatial fusion, purple arrows denote upsampling, and green arrows indicate downsampling. During this fusion process, the integration of low-level features with high-level features facilitates semantic information fusion [21,22]. In the process of asymptotic feature pyramid network, adaptive spatial fusion [23,24] is mainly used for multi-layer feature fusion. As shown in Equations (6) and (7), ASF (Adaptive Spatial Fusion) allocates spatial weights to enhance the saliency of key layers and reduce conflicting information between objects. $y_{ij}^l$ represents the result feature vector, $\alpha_{ij}^l$, $\beta_{ij}^l$ and $y_{ij}^l$ represent the spatial weights of the three levels at the l level, respectively, and the adaptive spatial fusion of multi-level features is obtained. $x_{ij}^{n \to l}$ represents the feature vector from n-level to l-level, and the position is $(i, j)$. In this way, the adaptive spatial fusion operation can effectively fuse feature vectors from different levels at each spatial position $(i, j)$ to enhance the effect of feature extraction.

$$y_{ij}^l = \alpha_{ij}^l * x_{ij}^{1 \to l} + \beta_{ij}^l * x_{ij}^{2 \to l} + \gamma_{ij}^l * x_{ij}^{3 \to l} \tag{6}$$

$$\alpha_{ij}^l + \beta_{ij}^l + \gamma_{ij}^l = 1 \tag{7}$$

**Inner-IoU improvement**

While the YOLOv10 network's loss function performs well in bounding box regression, it has limitations in adaptive adjustment and weak generalization when applied to different tea quality grades. In order to improve the detection performance of the YOLOv10 network and accelerate the bounding box regression, while enhancing the generalization ability of the model, this study further introduces Inner-IoU to optimize the YOLOv10 network, as shown in Fig 6. The Target Box and Anchor Box are represented in blue, while the Inner Target Box and Inner Anchor Box are shown in green. Inner-IoU evaluates the overlap between the predicted box and the ground truth by calculating the ratio of the green region to the yellow region. The center points of Inner Target Box and Target Box are c and $(x_d^{gt}, y_d^{gt})$, respectively. The center points of Anchor Box and Inner Anchor Box are c and $(x_d, y_d)$, respectively. The width and height are w and h, respectively. The improved YOLOv10 network uses a small auxiliary bounding box to accelerate convergence for samples with high IoU, and a large auxiliary bounding box to compute the loss for samples with low IoU.

To further improve the model's generalization capability, this study introduces a scaling factor (ratio $\in$ [0.5, 1.5]) to regulate auxiliary bounding box dimensions and loss computation, as detailed in Equations (8–15). When the ratio is less

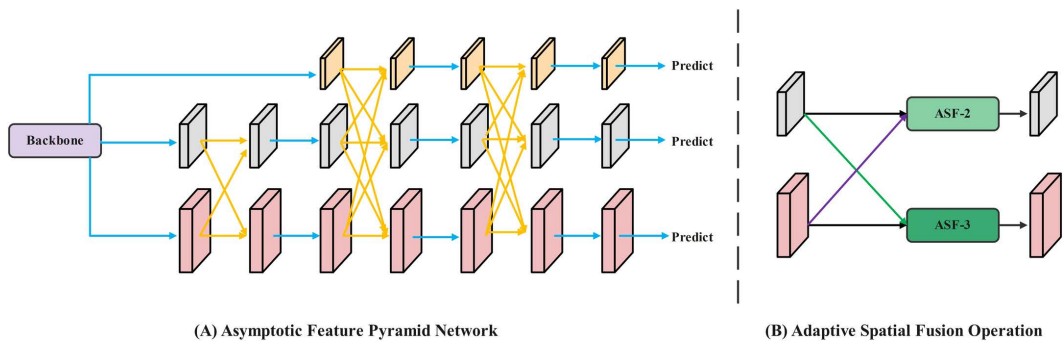

(A) Asymptotic Feature Pyramid Network     (B) Adaptive Spatial Fusion Operation

**Fig 5. Asymptotic feature pyramid network and adaptive spatial fusion.**

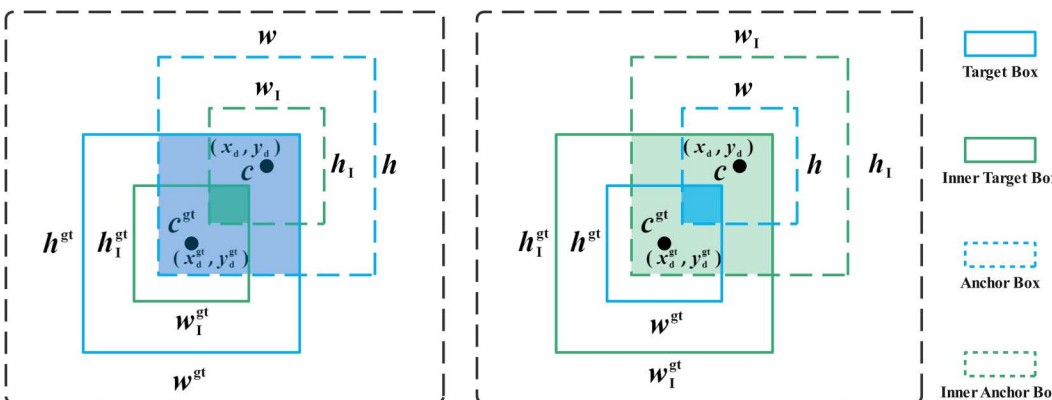

**Fig 6. Inner-IoU.**

than 1, the auxiliary bounding box becomes proportionally smaller than the ground truth box. Although the effective range for regression is smaller than that of IoU, the gradient's absolute value is higher than IoU, which accelerates the model's convergence. Which is highly beneficial for accelerating the model's convergence speed. When the ratio is greater than 1, the auxiliary bounding box can enhance the regression effect by expanding the effective range of regression.

$$L_{Inner-CIoU} = 1 + \frac{\rho^2\left(c, c^{gt}\right)}{d^2} + \alpha\nu - \frac{inter}{union} \tag{8}$$

$$inter = \left(\min\left(c_r^{gt}, c_r\right) - \max\left(c_l^{gt}, c_l\right)\right) * (\min\left(c_b^{gt}, c_b\right) - max(c_t^{gt}, c_t)) \tag{9}$$

$$union = \left(w^{gt} * h^{gt}\right) * (ratio)^2 + (w * h) * (ratio)^2 - inter \tag{10}$$

$$c_l^{gt} = x_d^{gt} - \frac{w^{gt} * ratio}{2}, \quad c_r^{gt} = x_d^{gt} + \frac{w^{gt} * ratio}{2} \tag{11}$$

$$c_t^{gt} = y_d^{gt} - \frac{h^{gt} * ratio}{2}, \quad c_b^{gt} = y_d^{gt} + \frac{h^{gt} * ratio}{2} \tag{12}$$

$$c_l = x_d - \frac{w * ratio}{2}, \quad c_r = x_d + \frac{w * ratio}{2} \tag{13}$$

$$c_t = y_d - \frac{h * ratio}{2}, \quad c_b = y_d + \frac{h * ratio}{2} \tag{14}$$

### Slicing aided hyper inference algorithm integration

Experimental results indicate that detection performance significantly declines beyond certain distance thresholds. For single buds and the "one bud, one leaf" configurations, both accuracy and recall rates decrease sharply when targets exceed 40 cm from the tea-picking robot. In contrast, the "one bud, two leaves" configuration shows similar decline beyond 60 cm. In the detection process for tea garden picking robots, accurately identifying and positioning long-distance and small targets can be challenging when relying solely on enhancements to the YOLOv10 network. To address this, this study combines the improved YOLOv10 network with the Slicing-aided Hyper Inference algorithm [25,26]. This combination aims

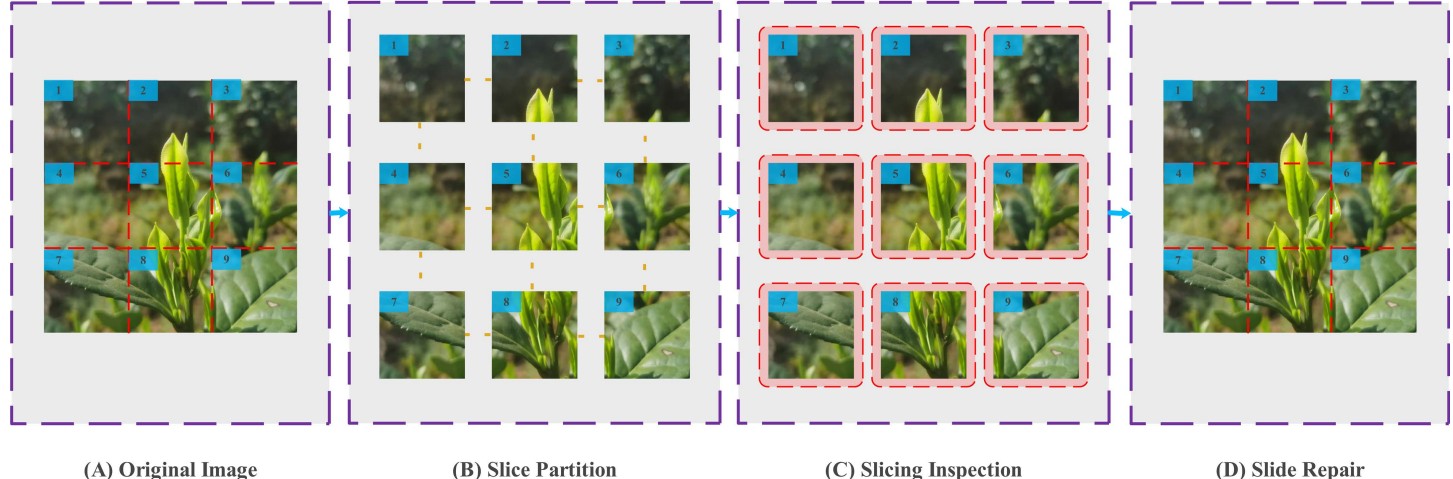

| (A) Original Image | (B) Slice Partition | (C) Slicing Inspection | (D) Slide Repair |

**Fig 7. Slicing aided hyper inference algorithm.**

to enhance the visual recognition model's ability to accurately detect and identify both long-distance and small targets. The Slicing Aided Hyper Inference Algorithm is integrated at the image input stage. As shown in Fig 7, the algorithm first divides the input image or video into multiple slices, with the number of slices customizable. It then performs target detection on each slice individually. The results are then merged back into the original image. During the slicing phase, the data is split into smaller chunks to reduce the computational and memory requirements for a single inference. In the hyper-inference phase, the algorithm processes each slice in parallel, significantly speeding up the overall operation. Finally, in the result fusion phase, the inference results from all slices are combined to ensure the accuracy of the final output. For overlapping detection boxes, the algorithm automatically merges them during the stitching process to enhance the accuracy of the recognition results. Additionally, by incorporating the Slicing-Aided Hyper Inference method, the model optimizes memory usage through image decomposition, significantly reducing hardware resource demands.

## Comparison experimental design

To validate the detection performance of the enhanced YOLOv10 network, which integrates Space-to-Depth Convolution, Asymptotic Feature Pyramid Network, and Inner-IoU, this study designed a comprehensive experimental framework. The framework includes four comparative test groups focused on Anji white tea's single buds, "one bud, one leaf," and "one bud, two leaves" configurations. The experiments involved training and testing models using the improved YOLOv10 network, YOLOv10, RT-DETR, CornerNet, and SSD networks on the dataset. Model training was conducted on a Windows 10 system with a GPU. The host machine featured an i5 12600KF processor (10 cores, 16 threads), a Colorful B760M-T WIFI V20 motherboard, a 512GB Colorful CN600 M.2 NVMe SSD, and a Colorful RTX4060TI ULTRA W OC 16GB graphics card. The network development was carried out using Python 3.9 and Pycharm 2023. The number of worker threads for data loading was set to 8, the batch size was set to 64, and the input image size was uniformly and automatically adjusted to 640×640 pixels. The training process was set for 1000 epochs.

As shown in Equations (15) and (17), P represents Precision, R represents Recall, and F1 function represents the harmonic mean of Precision and Recall, where FP represents the number of recognition errors, and FN represents the number of undetected. mAP represents the average accuracy, and FLOPs represents the number of floating-point operations. Precision serves as a key metric for assessing a model's predictive accuracy regarding positive class identification. The higher the Precision, the less the model misclassifies negative classes as positive classes. Recall is generally used

to evaluate the model 's ability to cover positive samples. The higher the Recall, the lower the false negative rate of the model. F1 is the value after considering the accuracy and recall rate. When there is a big difference between the accuracy and recall rate, F1 can better measure the comprehensive performance of the model.

$$P = \frac{T_P}{T_P + F_P} \tag{15}$$

$$R = \frac{T_P}{T_P + F_N} \tag{16}$$

$$F1 = 2 * \frac{Precision * Recall}{Precision + Recall}$$

## Results and analysis

### Ablation study

To assess the contribution of each component in the enhanced YOLOv10 network, an ablation study was performed. Various versions of the network were tested by removing one component at a time and measuring the resulting performance differences. The test results are presented in Table 2.

The results indicate that the Asymptotic Feature Pyramid Network enhances the YOLOv10 network by improving Precision by 1.75%, Recall by 2.58%, and mAP by 2.45%. Additionally, the Space-to-Depth Convolution boosts Precision by 3.63%, Recall by 0.75%, and mAP by 3.18%. The Inner-IoU technique increases Precision by 2.32%, Recall by 0.45%, and mAP by 2.29%. After the overall improvement, the Precision, Re-call and mAP of the YOLOv10-ASI network are increased by 7.1%, 6.69% and 6.78% respectively compared with the original YOLOv10 network. Compared to the 94.52% mAP value of our team's previous ShuffleNetv2-YOLOv5-Lite-E network, it shows an improvement of 2.4%. In order to further comprehensively evaluate the performance of the model, this study introduces the Gradient-weighted Class Activation Mapping heat map to visualize the model and judge the area of concern of the model. The GradCAM (Gradient-weighted Class Activation Mapping) heat map related to the ablation experiment is shown in Fig 8.

### Model analysis

In machine learning and deep learning, Loss Function [27,28] is mainly used to measure the difference between the model detection value and the true value. The smaller the value, the closer the model detection result is to the target value. In the YOLOv10 network, the Loss Function primarily consists of Classification Loss, Bounding Box Regression

**Table 2. Comparison results of ablation experiments.**

| Model | Precision(%) | Recall(%) | mAP(%) | Layers | Parameters | Gradients | GFLOPs |
|---|---|---|---|---|---|---|---|
| YOLOv10 | 84.86 | 89.33 | 91.14 | 402 | 2497778 | 2497762 | 8.2 |
| YOLOv10-A | 86.61 | 91.91 | 93.59 | 510 | 2913159 | 2913143 | 9.4 |
| YOLOv10-S | 88.49 | 90.08 | 94.32 | 379 | 11376962 | 11376946 | 40.8 |
| YOLOv10-I | 87.18 | 89.78 | 93.43 | 402 | 2497778 | 2497762 | 8.2 |
| YOLOv10-AS | 88.28 | 93.06 | 95.96 | 504 | 12111991 | 12111975 | 44.5 |
| YOLOv10-AI | 87.54 | 92.24 | 94.57 | 510 | 2913159 | 2913143 | 9.4 |
| YOLOv10-SI | 88.16 | 92.72 | 95.23 | 379 | 11376962 | 11376946 | 40.8 |
| **YOLOv10-ASI** | **91.96** | **96.02** | **97.92** | **504** | **12111991** | **12111975** | **44.5** |

Note: A represents the Asymptotic Feature Pyramid Network improvement; S represents Space-to-Depth Convolution; I represents Inner-IoU improvement.

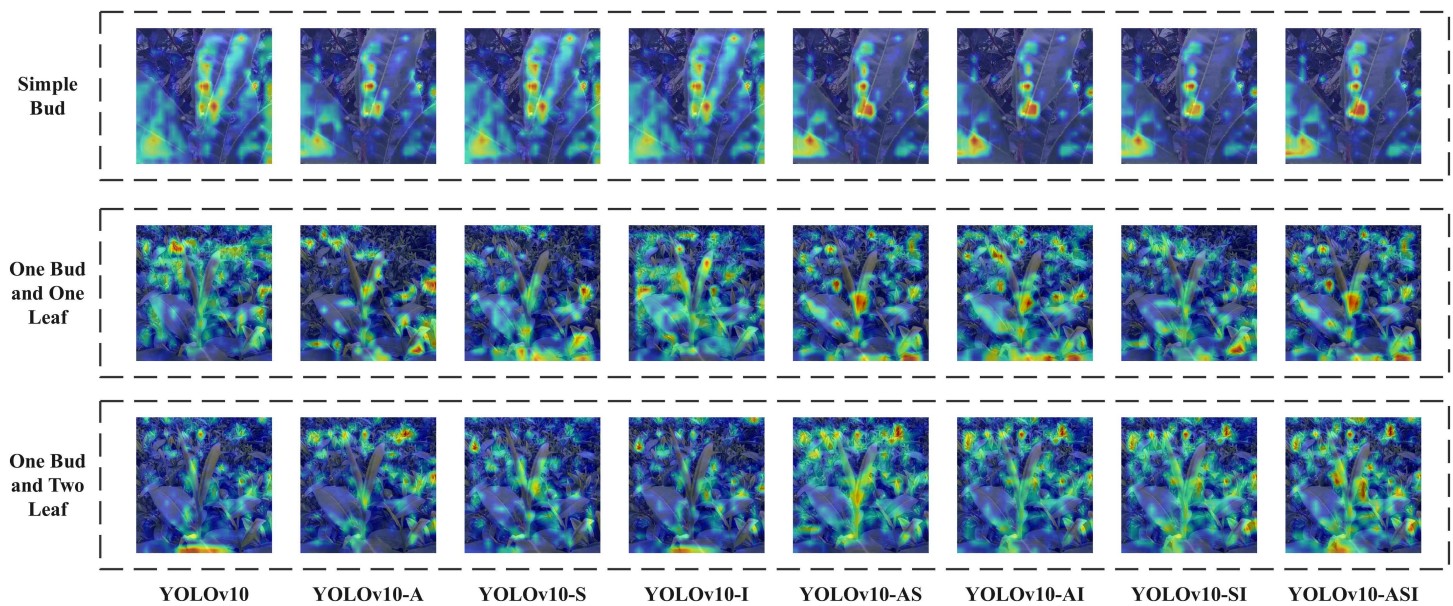

**Fig 8. Ablation Experiment GradCAM Heatmap.**

Loss, and Distribution Focal Loss. Classification Loss evaluates the model's accuracy in the classification task, while Bounding Box Regression Loss assesses the positional discrepancy between the predicted bounding box and the actual one. Distribution Focal Loss measures the divergence between the predicted distribution and the true distribution. As shown in Fig 9, the loss function of the YOLOv10-ASI network is significantly slowed down around 500 rounds, which is about 200 rounds earlier than the original YOLOv10 network. After 800 training rounds, the loss function of the YOLOv10-ASI network stabilized, achieving this milestone approximately 100 rounds earlier than the original YOLOv10 network. Specifically, Classification Loss, Bounding Box Regression Loss, and Distribution Focal Loss were ultimately maintained below 0.4, 0.4, and 0.9, respectively. Classification Loss and Distribution Focal Loss showed minimal variation in the training set compared to the original YOLOv10, while Bounding Box Regression Loss declined by over 30%. In the validation set, Classification Loss and Bounding Box Regression Loss each decreased by more than 60%, and Distribution Focal Loss experienced a reduction of around 10%.

Precision [29], Recall [30,31], and F1 [32,33], are commonly used to evaluate machine learning models and measure the performance of classification models. As shown in Fig 10, the YOLOv10-ASI network demonstrates a notable enhancement in Precision, Recall, and F1-Score compared to the original YOLOv10 network. Specifically, the improved YOLOv10 network boosts accuracy by 7.1 percentage points, recall by 6.69 percentage points, and the F1 by 6.78 percentage points.

## Model comparison experiment

To comprehensively evaluate the performance advantages of the proposed YOLOv10-ASI network, this study conducted model training and testing using the same dataset for the YOLOv10 network, YOLOv10, RT-DETR, CornerNet, and SSD networks. The test results are presented in Table 3. The findings indicate that the improved YOLOv10-ASI outperformed YOLOv10, RT-DETR, CornerNet, and SSD in single bud recognition, with AP values increasing by 6.10%, 8.47%, 23.70%, and 19.97%, respectively. For the recognition of one bud and one leaf, the AP values rose by 7.99%, 11.71%, 25.48%, and 19.85%, respectively. When recognizing one bud and two leaves, the AP values improved by 8.28%, 10.06%, 23.65%, and 20.76%, respectively. Overall, the mAP increased by 7.44%, 10.06%, 24.28%, and 20.19%, respectively.

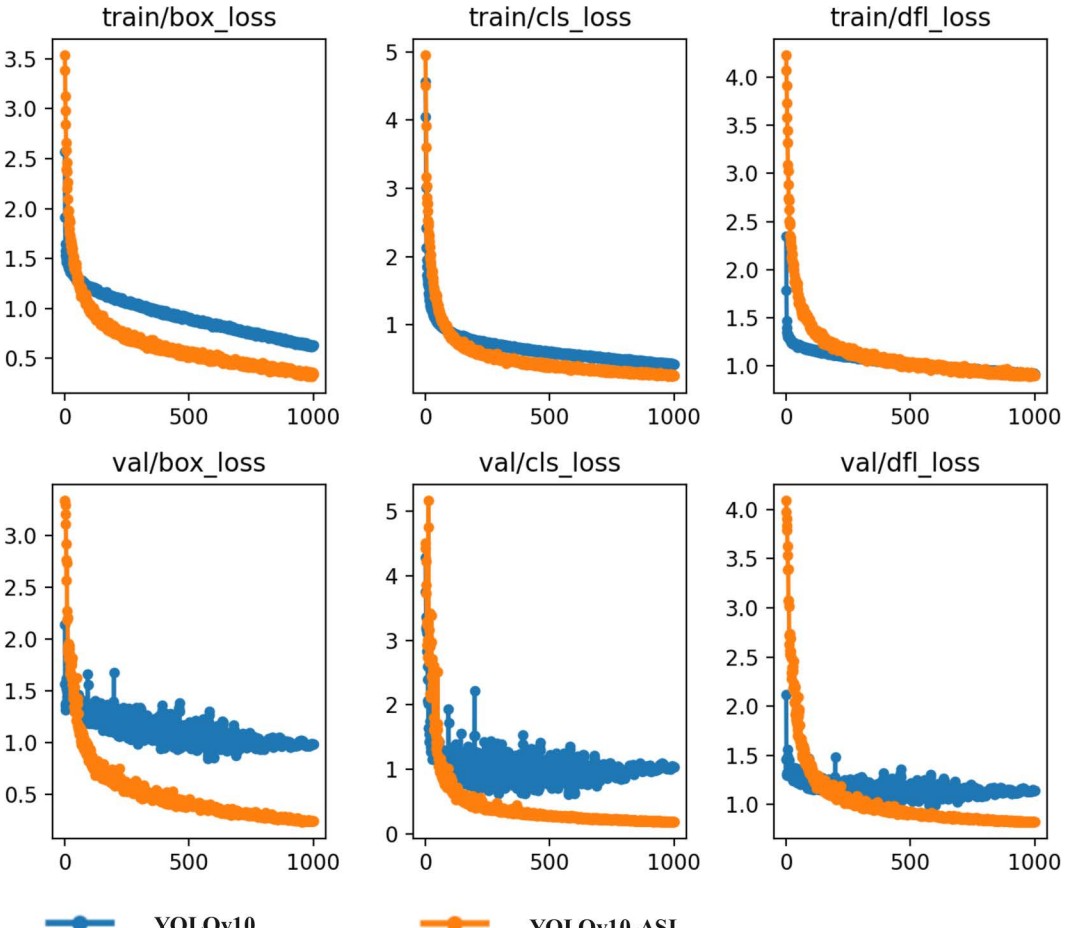

**Fig 9. Loss function variation curve.**

Additionally, in order to network combined with Slicing Aided Hyper Inference algorithm in abnormal conditions such as long distance, small target, low illumination and blur, this study performs additional external verification experiments. The external validation data were collected from the tea garden in Anji County, Huzhou City, Zhejiang Province, and some of the external validation results are shown in Fig 11. Experimental results demonstrate that both YOLOv10-ASI and the original YOLOv10 achieve the highest detection confidence among compared models, with YOLOv10-ASI exhibiting an average confidence score 5% higher than its baseline counterpart. In addition, for the long-distance and small target objects that cannot be detected by the YOLOv10-ASI network, the S-YOLOv10-ASI network after fusing the Slicing Aided Hyper Inference algorithm shows better detection results.

## Discussion

In recent years, the aging rural population and the migration of young people to urban areas have led to a growing scarcity of labor resources for tea picking. Additionally, the tea picking process requires a high level of selectivity for the buds, and manual picking [7,34] can be impacted by factors such as fatigue or inexperience, which can compromise the quality of the tea. With the advancement of agricultural automation and intelligence, the development and application of tea picking robots have become particularly important. Aiming at the challenges of long distance, small target and low resolution

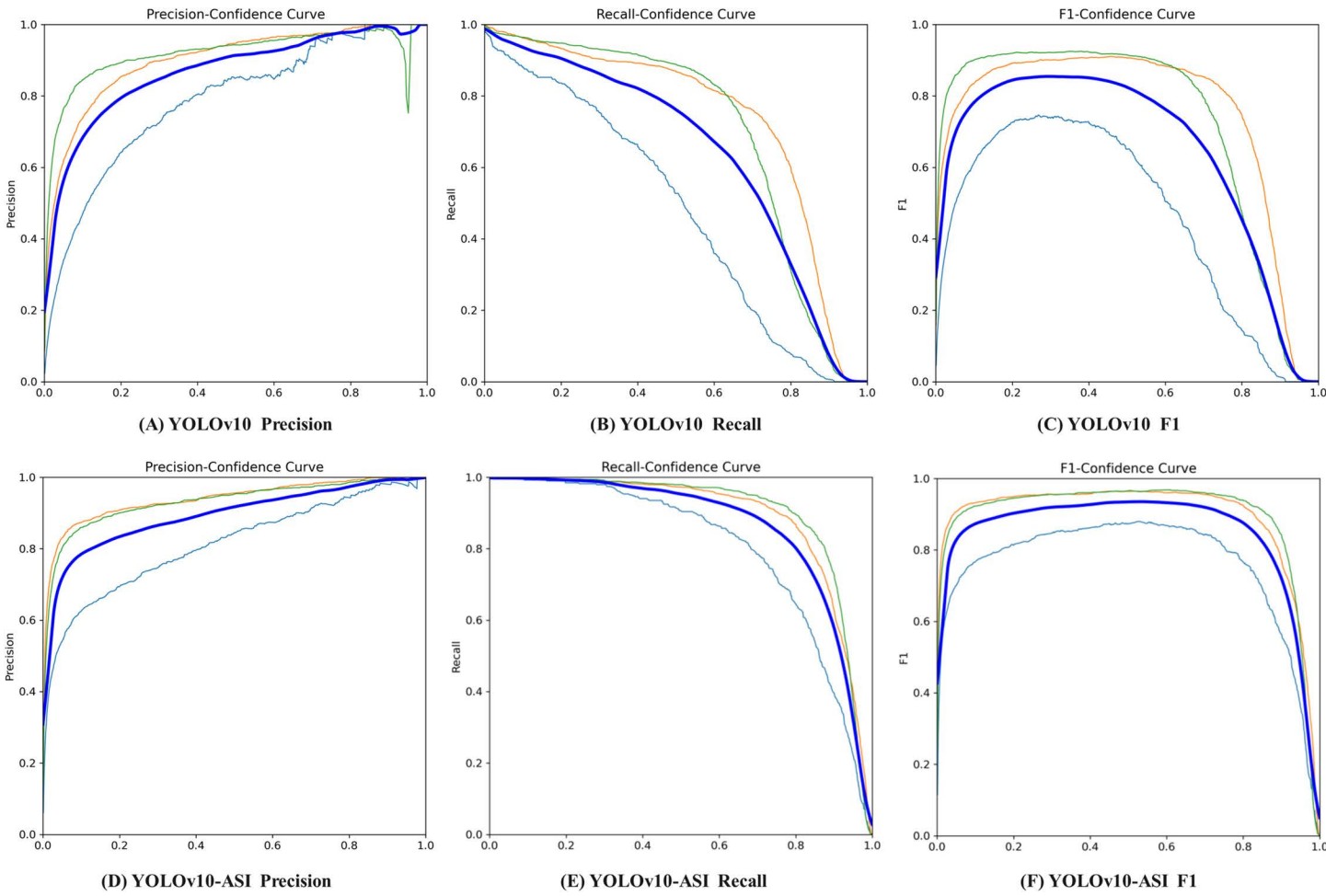

**Fig 10. Precision, recall, F1-Score curve.**

**Table 3. Model comparison experimental results.**

| Model | AP(Simple Bud) | AP(One Bud One Leave) | AP(One Bud Two Leaves) | Precision(%) | Recall(%) | mAP(%) |
|---|---|---|---|---|---|---|
| YOLOv10-ASI | **98.65** | **98.48** | **96.63** | **91.96** | **96.02** | **97.92** |
| YOLOv10 | 92.98 | 91.19 | 89.24 | 84.86 | 89.33 | 91.14 |
| RT-DETR | 90.95 | 88.16 | 87.80 | 82.01 | 85.83 | 88.97 |
| CornerNet | 79.75 | 78.48 | 78.15 | 78.36 | 73.75 | 78.79 |
| SSD | 82.23 | 82.17 | 80.02 | 81.31 | 75.38 | 81.47 |

encountered by traditional tea picking robots, this study uses Space-to-Depth Convolution to improve some structures of YOLOv10 network to reduce the loss of detail information of long distance and low resolution images.

The proposed framework incorporates an Asymptotic Feature Pyramid Network to address several limitations in YOLOv10: information attenuation across network stages, insufficient key feature emphasis, object feature conflicts, and suboptimal distant-layer feature fusion. At the same time, the Inner-IoU optimization loss function is used to improve the convergence speed of the model and enhance the universality of the model. Combined with the Slicing Aided Hyper

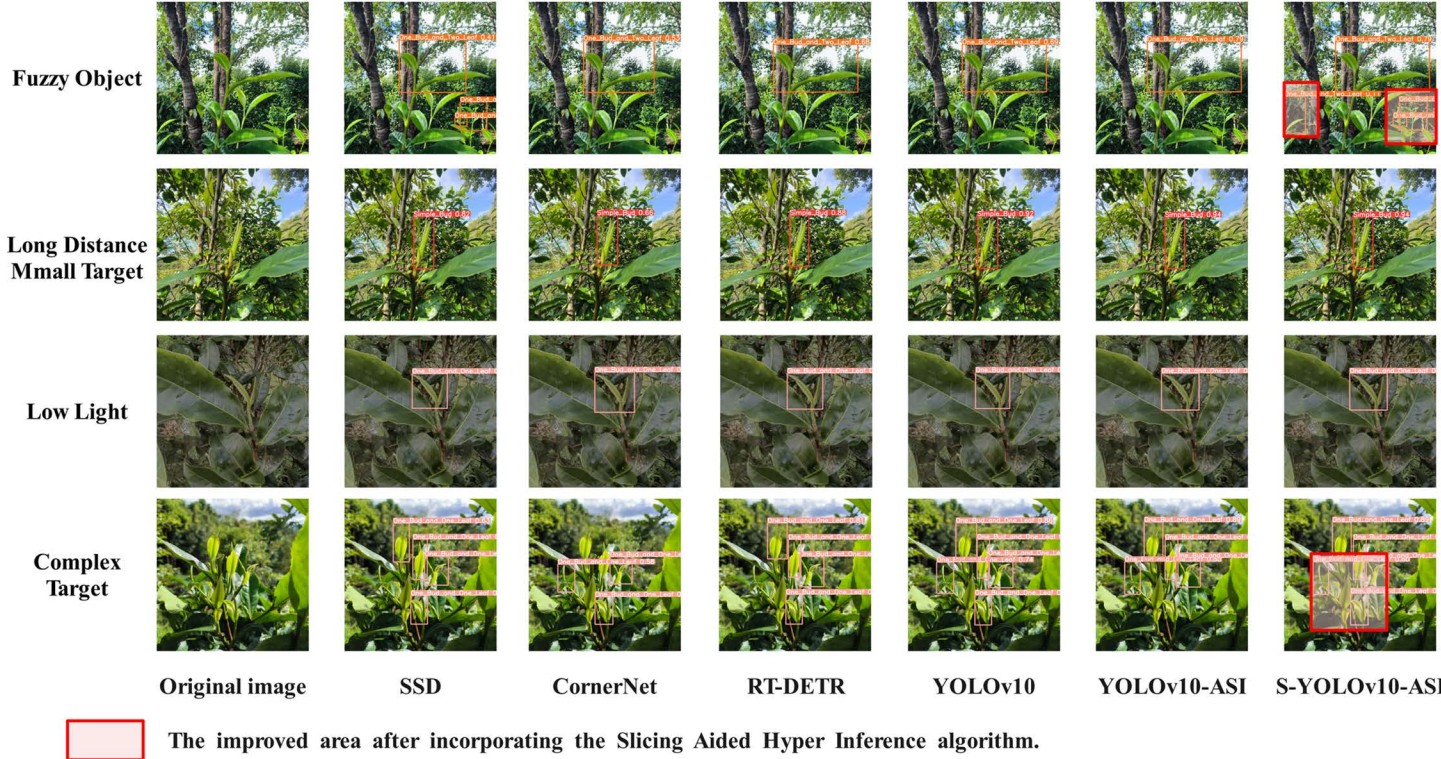

| Original image | SSD | CornerNet | RT-DETR | YOLOv10 | YOLOv10-ASI | S-YOLOv10-ASI |

Rows (left labels): Fuzzy Object, Long Distance Mmall Target, Low Light, Complex Target

☐ The improved area after incorporating the Slicing Aided Hyper Inference algorithm.

**Fig 11. Different model detection results comparison.**

Inference algorithm [35,36], the performance in the detection of long-distance and small targets is improved. The visual recognition model for the Anji white tea picking robot developed in this study offers robust technical support for the intelligent harvesting of Anji white tea, demonstrating significant application potential and value for promotion.

Furthermore, the proposed models and methodologies exhibit substantial potential for broad applications. Beyond facilitating the harvesting of diverse tea varieties, these techniques can be readily adapted for intelligent crop harvesting across agricultural domains. This research holds considerable theoretical and practical value in advancing the intelligent modernization of agricultural production [37,38] and achieving sustainable agricultural development.

## Conclusion

In this study, the partial structure of YOLOv10 network is improved by Space-to-Depth Convolution, which reduces the loss of detail information of long-distance and low-resolution images. The YOLOv10 network framework is optimized by the asymptotic feature pyramid network, which enhances the saliency of the key layer, alleviates the contradictory information of different objects, and optimizes the fusion of non-adjacent layers. Through the calculation of Inner-IoU optimization loss function, the convergence speed of the model is improved, and the universality of the model is further enhanced. After fusing the slice-assisted hyper-inference algorithm, the S-YOLOv10-ASI network shows better detection results for long-distance and small target objects that cannot be detected by the YOLOv10-ASI network. The experimental results show that:

(1) From the perspective of loss function trends, the Classification Loss, Bounding Box Regression Loss, and Distribution Focal Loss of the S-YOLOv10-ASI network eventually stabilized at values below 0.4, 0.4, and 0.9, respectively. While the Classification Loss and Distribution Focal Loss showed no significant changes in the training set compared

to the original YOLOv10 network, the Bounding Box Regression Loss decreased by more than 30%. In the validation set, the Classification Loss and Bounding Box Regression Loss decreased by more than 60%, while the Distribution Focal Loss decreased by approximately 10%. This indicates a noticeable improvement in model stability. Additionally, the loss function of the YOLOv10-ASI network begins to stabilize significantly around 500 iterations, which is approximately 200 iterations earlier than in the original YOLOv10 network.

(2) Model Parameters and Detection Performance: Compared to YOLOv10, RT-DETR, CornerNet, and SSD, the improved S-YOLOv10-ASI network showed AP improvements of 6.10%, 8.47%, 23.70%, and 19.97% for single bud recognition, respectively. For one bud with one leaf recognition, AP values improved by 7.99%, 11.71%, 25.48%, and 19.85%, respectively. For one bud with two leaves recognition, AP values improved by 8.28%, 10.06%, 23.65%, and 20.76%, respectively. The final mAP im-proved by 7.44%, 10.06%, 24.28%, and 20.19%, respectively. Additionally, Precision increased by 8.37%, 12.13%, 17.36%, and 13.10%, while Recall increased by 7.49%, 11.87%, 30.20%, and 27.38%. The S-YOLOv10-ASI network also demonstrated superior detection performance for long-distance and small-target objects that traditional networks could not detect.

## Supporting information

**S1 File. Striking image.**
(TIFF)

## Author contributions

**Conceptualization:** Chunhua Yang.

**Data curation:** Wenxia Yuan.

**Formal analysis:** Zejun Wang.

**Funding acquisition:** Baijuan Wang.

**Investigation:** Bowu Song.

**Methodology:** Zejun Wang.

**Project administration:** Xianqiu Dong.

**Resources:** Yuandong Xiao.

**Software:** Qiang Zhao, Shihao Zhang.

**Validation:** Qiang Zhao, Shihao Zhang.

**Visualization:** Qiang Zhao, Shihao Zhang.

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
