## [Decision Letter · Decision Letter 0]

PONE-D-24-60012Identification of Fresh Leaves of Anji White Tea : S-YOLOv10-SIC Algorithm Fusing Asymptotic Feature Pyra-mid NetworkPLOS ONE

Dear Dr. Wang,

Thank you for submitting your manuscript to PLOS ONE. After careful consideration, we feel that it has merit but does not fully meet PLOS ONE’s publication criteria as it currently stands. Therefore, we invite you to submit a revised version of the manuscript that addresses the points raised during the review process.

We look forward to receiving your revised manuscript.

Kind regards,

Yile Chen, Ph.D. in Architecture

Academic Editor

PLOS ONE

Journal Requirements:

2. Please note that PLOS ONE has specific guidelines on code sharing for submissions in which author-generated code underpins the findings in the manuscript. In these cases, we expect all author-generated code to be made available without restrictions upon publication of the work. Please review our guidelines at https://journals.plos.org/plosone/s/materials-and-software-sharing#loc-sharing-code and ensure that your code is shared in a way that follows best practice and facilitates reproducibility and reuse

6. Please include a separate caption for each figure in your manuscript

Reviewers' comments:

Reviewer's Responses to Questions

**Comments to the Author**

1. Is the manuscript technically sound, and do the data support the conclusions?

Reviewer #1: Partly

Reviewer #2: Partly

2. Has the statistical analysis been performed appropriately and rigorously? 

Reviewer #1: N/A

Reviewer #2: No

3. Have the authors made all data underlying the findings in their manuscript fully available?

Reviewer #1: No

Reviewer #2: Yes

4. Is the manuscript presented in an intelligible fashion and written in standard English?

Reviewer #1: Yes

Reviewer #2: No

5. Review Comments to the Author

Reviewer #1: 1- One of the challenges of tea picking is the density of leaves. While according to the authors’ claims in the article, like the abstract, the proposal model has been applied for selective pictures and conditions; for instance, in the abstract, they said, "The AP values for single bud, one bud and one leaf, and one bud and two leaves have seen improvements of 6.10%, 7.99%, and 8.28%.". These results seem to be cross-sectional and ad hoc. It seems that the proposed model was tested and trained on images with specific and determined conditions. This means that the conditions of the images are laboratory and ad hoc and cannot guarantee the same result in natural conditions among the crowding of leaves and buds. What is the authors’ explanation?

2- The correct definition of the long-distance and small-target parameters is not provided. According to what the authors have presented, they have emphasized more on light and brightness conditions. How and where are these two parameters applied in the model?

3- The model is not properly and clearly described and is not understandable to readers, especially those who are not familiar with the methods and algorithms. The authors' explanations to the reviewers, although commendable, should be understood by the readers.

4- Pictures in this paper have low quality. And why don't the pictures have numbers?

5- The article's text has repetitions in both sentences and content.

Reviewer #2: 1. Is the manuscript technically sound, and do the data support the conclusions?

Answer: The manuscript is technically sound and generally presents data to support the conclusions. However, several areas require further improvement:

Statistical Analysis : The absence of formal statistical testing, such as through p-values or confidence intervals, reduces the strength associated with the claims. The inclusion of testing for significance would enhance the validity of the results.

Methodology Clarity: Major changes, such as "merging of nonadjacent layers," and the introduction of the Slicing Aided Hyper Inference algorithm, should be better explained in order to be reproducible.

Experimental Rigor: While the dataset is well-described, certain augmentation parameters are not justified; for example, changing brightness and contrast. In addition, the timing metrics are not reported-that is, inference speed-which makes an assessment of practical feasibility difficult.

Conclusions: Some claims, such as information loss reduction and convergence speed improvement, are not quantitatively validated and will be supported with specific data.

Addressing such gaps will ensure that the technical rigors within the manuscript are well-matched with the conclusion.

2. Whether the statistical analysis is proper and rigorous?

Answer: The statistical analysis is insufficiently rigorous in the current manuscript and requires essential and key improvements. Key concerns include:

Lack of Formal Statistical Testing: The manuscript does not present the p-values, confidence intervals, or other measures that would confirm the statistical significance of the reported improvement in metrics such as Precision, Recall, and mAP 4. Such an analysis is absolutely necessary to be included in the manuscript to prove the strength of the findings.

Outliers Not Explained : The loss function of YOLOv10 in Figure 9 for the second validation set has an outlier at 40. This was not explained and analyzed, raising some doubts in the results.

Lack of Justification of Parameters: Justification of the parameters of data augmentation, like adjustments in brightness and contrast, is based on empirical evidence or references.

Conclusions: The manuscript requires formal statistical testing, a clearer explanation of the outliers, and justification of the experimental choices.

4.Is the manuscript presented in an intelligible fashion and written in standard English?

Answer: The manuscript is not in a readable state and suffers from numerous problems in grammar, clarity, and format that interfere with reading.

Grammatical Errors and Awkward Phrasing: Most of the sentences are either very long and redundant or poorly structured, making the text hard to follow. For example, the abstract is too long and not clear at all.

Repetition: Several statements, such as the improvement on Precision, Recall, and mAP, have been repeated several times without giving any further insight; the flow and depth of the manuscript were reduced accordingly.

Figures and Labels: Scattered all along the text, several figures-including Figure 3 (Improved YOLOv10 Network Structure) and Figure 7 (Slicing Aided Hyper Inference Algorithm)-are missing proper labeling or annotation, leading to much reader confusion.

Generally, careful editing is needed regarding typos, language use, and overall expression.

Conclusions: The manuscript requires professional English language editing in order to increase clarity, proper sentence structure, and quality in figures, since this paper still needs significant changes to meet PLOS ONE requirements.

6. PLOS authors have the option to publish the peer review history of their article (what does this mean? ). If published, this will include your full peer review and any attached files.

**Do you want your identity to be public for this peer review?** For information about this choice, including consent withdrawal, please see our Privacy Policy .

Reviewer #1: **Yes: ** Somayeh Mehrabadi

Reviewer #2: **Yes: ** Md. Abdullah Al Mashud

---

## [Author Response · Author response to Decision Letter 1]

4 Mar 2025

Thanks very much for your time to review this manuscript. I really appreciate your comments and suggestions. We have considered these comments carefully and tried our best to address every one of them. And the corresponding part in the text has been modified using red font.

Review 1:

1. One of the challenges of tea picking is the density of leaves. While according to the authors’ claims in the article, like the abstract, the proposal model has been applied for selective pictures and conditions; for instance, in the abstract, they said, "The AP values for single bud, one bud and one leaf, and one bud and two leaves have seen improvements of 6.10%, 7.99%, and 8.28%.". These results seem to be cross-sectional and ad hoc. It seems that the proposed model was tested and trained on images with specific and determined conditions. This means that the conditions of the images are laboratory and ad hoc and cannot guarantee the same result in natural conditions among the crowding of leaves and buds. What is the authors’ explanation?

Modification instructions: Thank you very much for your valuable suggestions. To enhance the model's detection performance in real-world applications, we simulated the distance between the tea-picking robot and the tea plants in our experiments. Additionally, to ensure the model can accurately recognize tea leaves under various lighting conditions, slopes, and angles, and improve its generalizability in practical applications, we adopted a multi-stage, multi-angle image collection approach. These measures help strengthen the model's adaptability, particularly when dealing with the complex and variable conditions found in tea gardens. We are also well aware that real-world conditions are far more challenging than laboratory settings, especially considering the complexity and dynamic nature of tea plant growth and distribution in tea gardens. To address this, we further improved the model's generalization ability through image augmentation techniques, simulating variations in lighting, perspectives, and distances to better equip the model for handling various uncertainties in natural environments. To further validate the model's performance in non-ideal conditions, we not only carefully designed the training and validation sets during dataset splitting, but also specially extracted a portion of samples as a test set for external validation (as shown in Figure 11). These efforts ensure that our proposed model has strong practical applicability and can operate stably in complex environments. Once again, thank you for your thorough review and valuable feedback. We will continue to make improvements to ensure the model's reliability and practicality.

2. The correct definition of the long-distance and small-target parameters is not provided. According to what the authors have presented, they have emphasized more on light and brightness conditions. How and where are these two parameters applied in the model?

Modification instructions: Have been changed in accordance with the advice given. Thank you very much for your valuable feedback. Regarding the issue you raised about the definitions of 'long distance' and 'small target,' we would like to clarify further. In this study, we indeed focused on lighting and brightness conditions to enhance the model's robustness in different environments. However, 'long distance' and 'small target' represent another key challenge we are addressing, particularly in the real-world scenarios of tea-picking robot applications. Through testing, we found that when the target is more than 40 cm away from the tea-picking robot, the model's precision and recall for detecting small buds and the 'one bud, one leaf' configuration start to decrease rapidly. However, for the 'one bud, two leaves' configuration, precision and recall only begin to decline significantly when the target is over 60 cm away. Furthermore, in terms of target size, tea buds and leaves are usually quite small, especially when they are at a distance or when image resolution is low. Therefore, we define 'small targets' as those tea leaves that are small in size and occupy a relatively small number of pixels in the image. The improvement in this area is reflected not only in the model's Precision, Recall, and mAP but also in the external experiments shown in Figure 11. Once again, thank you for your detailed review and valuable suggestions.

3. The model is not properly and clearly described and is not understandable to readers, especially those who are not familiar with the methods and algorithms. The authors' explanations to the reviewers, although commendable, should be understood by the readers.

Modification instructions: Have been changed in accordance with the advice given. Thank you very much for your valuable feedback. We greatly appreciate your comments regarding the unclear description of the model, and we sincerely accept your suggestions. We understand that for readers who are not familiar with the methods and algorithms we used, the model description may indeed be somewhat difficult to follow. To address this, we have made further revisions and improvements to the paper, aiming to make it clearer and more accessible, especially for those who may not be familiar with the field.

4. Pictures in this paper have low quality. And why don't the pictures have numbers?

Modification instructions: Thank you very much for your valuable feedback. Regarding the image quality issue you mentioned, we did upload high-resolution images to the system. However, due to file size limitations, the system may have automatically compressed the images when transmitting them to the reviewers, which may have caused a decrease in quality. We sincerely apologize for this. As for the image numbering, we followed the editor's instructions and named the image files according to the corresponding numbers during the upload process. We understand that both image clarity and numbering are crucial for review and reading. Therefore, if you require higher-resolution original images, we would be happy to provide them at any time, either by sending them to the email address or other contact details you provide. Once again, thank you for your thorough review and valuable suggestions.

5. The article's text has repetitions in both sentences and content.

Modification instructions: Have been changed in accordance with the advice given. Thank you very much for your valuable feedback. We have carefully reviewed the paper and made further revisions to address the repetitive sentences and content. We appreciate your thoughtful review and feedback.

Review 2:

1. Is the manuscript technically sound, and do the data support the conclusions? Answer: The manuscript is technically sound and generally presents data to support the conclusions. However, several areas require further improvement: Statistical Analysis : The absence of formal statistical testing, such as through p-values or confidence intervals, reduces the strength associated with the claims. The inclusion of testing for significance would enhance the validity of the results. Methodology Clarity: Major changes, such as "merging of nonadjacent layers," and the introduction of the Slicing Aided Hyper Inference algorithm, should be better explained in order to be reproducible. Experimental Rigor: While the dataset is well-described, certain augmentation parameters are not justified; for example, changing brightness and contrast. In addition, the timing metrics are not reported-that is, inference speed-which makes an assessment of practical feasibility difficult. Conclusions: Some claims, such as information loss reduction and convergence speed improvement, are not quantitatively validated and will be supported with specific data. Addressing such gaps will ensure that the technical rigors within the manuscript are well-matched with the conclusion.

Modification instructions: Thank you very much for your valuable suggestions. Regarding the statistical analysis issue you raised, after reviewing a large number of relevant literatures, especially in the research on the YOLO series networks, we found that for evaluating the performance of visual recognition networks, metrics such as precision, recall, and mean average precision (mAP) are commonly used, rather than traditional statistical tests (e.g., p-values or confidence intervals). These metrics are widely accepted and applied in the field for evaluating object detection algorithms' performance. Additionally, the output of visual recognition networks is primarily based on these performance metrics, and statistical tests are not used as evaluation criteria for visual recognition detection models. Therefore, we have used these widely recognized evaluation metrics in our paper without conducting additional statistical analysis. Thank you for your suggestion, and we will further explore how to combine statistical tests in future research to enhance the validity of the results, thereby providing stronger support and evidence for the research.

Thank you very much for your valuable feedback. Regarding the issue you raised about the 'clarity of methodology,' particularly concerning the introduction of 'merging non-adjacent layers' and the 'Slicing Aided Hyper Inference Algorithm,' we have made further revisions to the paper to ensure these significant changes are presented more clearly, making the research more reproducible. The added content is: “To address the aforementioned issues, this study introduces an enhanced Asymptotic Feature Pyramid Network to optimize the YOLOv10 architecture. This improvement enables effective fusion between non-adjacent layers by progressively incorporating higher-level features. In the refined YOLOv10 network, the initial stage focuses on integrating low-level features, which are rich in detailed information. As the fusion process advances, higher-level features are incrementally introduced, as illustrated in Fig. 5. This gradual integration of higher-level features significantly reduces the semantic gap between non-adjacent layers, thereby facilitating more effective fusion of features across different levels.” “The Slicing Aided Hyper Inference Algorithm is strategically positioned between the input and the backbone during the model's detection phase.”

Thank you very much for your valuable feedback. Regarding the 'experimental rigor' issue you raised, in our study, variations in brightness and contrast were simulated based on common lighting changes and different shooting conditions in tea garden environments. To ensure the model's robustness in various environments, we made random adjustments to the brightness (ranging from 0.5 to 1.5 times) and contrast (ranging from 0.4 to 1.8 times) of the images during the experiments. This data augmentation strategy aims to simulate natural lighting variations, different shooting angles, and environmental factors in the tea garden. With these enhancements, our model is better equipped to handle uneven lighting, shadows, and reflections, thus improving its performance in real-world conditions. We have added a detailed explanation of these augmentation parameters in the paper. Regarding the inference speed issue you mentioned, our tests show that inference speed varies depending on the device and parameter settings, and there may be some fluctuation in inference time during each test round. Therefore, we believe that relying solely on inference speed may not accurately reflect the model's performance, especially when trying to replicate our experiments on different hardware. For this reason, we used more accurate and standardized comparison factors in this study, such as Parameters, Gradients, and GFLOPs, which better reflect the model's computational complexity and actual inference performance. By calculating the total number of parameters in the model, we can gain insight into the model's complexity. A larger number of parameters generally indicates greater computational demands and longer inference times. The gradient amount reflects the computational cost for each update during the training process. The size of the gradients is closely related to the model's complexity and inference speed, making it an important factor in evaluating model performance. GFLOPs, or Giga Floating Point Operations per Second, measures how many floating-point operations the model performs per second and is a key indicator of the model's inference speed and computational efficiency. A higher GFLOPs value indicates that the model requires more computational resources during inference, leading to a corresponding decrease in inference speed.

Thank you very much for your valuable feedback. Regarding the conclusion section, especially the issue you raised about the lack of quantitative validation for 'reduced information loss' and 'improved convergence speed,' we have revised the paper and, based on your suggestions, added specific data to support these points.

2. Lack of Formal Statistical Testing: The manuscript does not present the p-values, confidence intervals, or other measures that would confirm the statistical significance of the reported improvement in metrics such as Precision, Recall, and mAP 4. Such an analysis is absolutely necessary to be included in the manuscript to prove the strength of the findings. Outliers Not Explained : The loss function of YOLOv10 in Figure 9 for the second validation set has an outlier at 40. This was not explained and analyzed, raising some doubts in the results. Lack of Justification of Parameters: Justification of the parameters of data augmentation, like adjustments in brightness and contrast, is based on empirical evidence or references. Conclusions: The manuscript requires formal statistical testing, a clearer explanation of the outliers, and justification of the experimental choices.

Modification instructions: Thank you very much for your valuable suggestions. Regarding the statistical analysis issue you raised, after reviewing a large number of relevant literatures, especially in the research on the YOLO series networks, we found that for evaluating the performance of visual recognition networks, metrics such as precision, recall, and mean average precision (mAP) are commonly used, rather than traditional statistical tests (e.g., p-values or confidence intervals). These metrics are widely accepted and applied in the field for evaluating object detection algorithms' performance. Additionally, the output of visual recognition networks is primarily based on these performance metrics, and statistical tests are not used as evaluation criteria for visual recognition detection models. Therefore, we have used these widely recognized evaluation metrics in our paper without conducting additional statistical analysis. Thank you for your suggestion, and we will further explore how to combine statistical tests in future research to enhance the validity of the results, thereby providing stronger support and evidence for the research.

Thank you very much for your valuable feedback. Regarding the issue you raised about the outlier observed in the YOLOv10 loss function for the second validation set at around 40 iterations in Figure 9, during our training process, the model did not use pre-trained weights and was trained from scratch. This means that the model's initial parameters were quite random, and as a result, there may have been noticeable fluctuations in the loss function during the early stages of training. Particularly in the initial phase, when the network weights have not been effectively optimized, larger-than-usual fluctuations in the loss function can occasionally occur. This is a common phenomenon when training deep learning models from a random initialization. In the early stages of training visual recognition models, parameter adjustments are often more drastic, causing short-term fluctuations in the loss function. As the model gradually converges, the loss function stabilizes. These abnormal fluctuations do not affect the final model performance, and the fluctuations observed in the early training phase did not have a negative impact on the overall performance of the model.

Thank you very much for your valuable feedback. Regarding the issue of the rationality of the data augmentation parameters,

---

## [Decision Letter · Decision Letter 1]

PONE-D-24-60012R1Identification of Fresh Leaves of Anji White Tea : S-YOLOv10-ASI Algorithm Fusing Asymptotic Feature Pyra-mid NetworkPLOS ONE

Dear Dr. Wang,

Thank you for submitting your manuscript to PLOS ONE. After careful consideration, we feel that it has merit but does not fully meet PLOS ONE’s publication criteria as it currently stands. Therefore, we invite you to submit a revised version of the manuscript that addresses the points raised during the review process.

We look forward to receiving your revised manuscript.

Kind regards,

Yile Chen, Ph.D. in Architecture

Academic Editor

PLOS ONE

Journal Requirements:

Additional Editor Comments (if provided):

This article has been revised and improved. However, there may be some repeated expressions in some sentences. Please proofread and revise them carefully.

Reviewers' comments:

Reviewer's Responses to Questions

**Comments to the Author**

1. If the authors have adequately addressed your comments raised in a previous round of review and you feel that this manuscript is now acceptable for publication, you may indicate that here to bypass the “Comments to the Author” section, enter your conflict of interest statement in the “Confidential to Editor” section, and submit your "Accept" recommendation.

Reviewer #1: (No Response)

Reviewer #2: All comments have been addressed

2. Is the manuscript technically sound, and do the data support the conclusions?

Reviewer #1: (No Response)

Reviewer #2: Yes

3. Has the statistical analysis been performed appropriately and rigorously? 

Reviewer #1: (No Response)

Reviewer #2: Yes

4. Have the authors made all data underlying the findings in their manuscript fully available?

Reviewer #1: (No Response)

Reviewer #2: Yes

5. Is the manuscript presented in an intelligible fashion and written in standard English?

Reviewer #1: (No Response)

Reviewer #2: Yes

6. Review Comments to the Author

Reviewer #1: - The article's text has repetitions in both sentences and content.

- The model stil is not properly and clearly described and is not understandable to readers, especially those who are not familiar with the methods and algorithms. The authors' explanations to the reviewers, although commendable, should be understood by the readers.

Reviewer #2: (No Response)

7. PLOS authors have the option to publish the peer review history of their article (what does this mean? ). If published, this will include your full peer review and any attached files.

**Do you want your identity to be public for this peer review?** For information about this choice, including consent withdrawal, please see our Privacy Policy .

Reviewer #1: **Yes: ** Somayeh Mehrabadi

Reviewer #2: **Yes: ** Md. Abdullah Al Mashud

---

## [Author Response · Author response to Decision Letter 2]

4 Apr 2025

Thanks very much for your time to review this manuscript. I really appreciate your comments and suggestions. We have considered these comments carefully and tried our best to address every one of them. And the corresponding part in the text has been modified using red font.

Additional Editor Comments:

1. This article has been revised and improved. However, there may be some repeated expressions in some sentences. Please proofread and revise them carefully.

Modification instructions: Thank you for your valuable review comments and suggestions! We have made detailed revisions to the manuscript and carefully reviewed all sentences for any possible redundant expressions. All revisions are marked in the 'Revised Manuscript with Track Changes' section, with red font used for differentiation to facilitate your review. Once again, thank you for your help, and we look forward to your further feedback!

Reviewer 1:

2. The model stil is not properly and clearly described and is not understandable to readers, especially those who are not familiar with the methods and algorithms. The authors' explanations to the reviewers, although commendable, should be understood by the readers.

Modification instructions: Thank you for the valuable comments from you and the reviewers. We have made detailed revisions addressing the issue of redundancy in the manuscript as raised by the reviewers, ensuring that there are no unnecessary repetitions in the sentences and content. In addition, we have further clarified and improved the description of the model to ensure that even readers who are not familiar with the relevant methods and algorithms can easily understand. During the revision process, we have simplified the expression as much as possible to make it more comprehensible. All revisions are marked in the 'Revised Manuscript with Track Changes' section, with red font used to facilitate your and the reviewers' review.

---

## [Decision Letter · Decision Letter 2]

PONE-D-24-60012R2Identification of Fresh Leaves of Anji White Tea : S-YOLOv10-ASI Algorithm Fusing Asymptotic Feature Pyra-mid NetworkPLOS ONE

Dear Dr. Wang,

Thank you for submitting your manuscript to PLOS ONE. After careful consideration, we feel that it has merit but does not fully meet PLOS ONE’s publication criteria as it currently stands. Therefore, we invite you to submit a revised version of the manuscript that addresses the points raised during the review process.

We look forward to receiving your revised manuscript.

Kind regards,

Yile Chen, Ph.D. in Architecture

Academic Editor

PLOS ONE

Journal Requirements:

Additional Editor Comments :

The reviewers' questions have been basically addressed and accepted. There are still some repeated sentences, please check and delete them. At the same time, add the serial number name and description of the picture. After resolving these minor issues, I consider the article acceptable for publication.

Reviewers' comments:

Reviewer's Responses to Questions

**Comments to the Author**

1. If the authors have adequately addressed your comments raised in a previous round of review and you feel that this manuscript is now acceptable for publication, you may indicate that here to bypass the “Comments to the Author” section, enter your conflict of interest statement in the “Confidential to Editor” section, and submit your "Accept" recommendation.

Reviewer #1: (No Response)

Reviewer #3: All comments have been addressed

2. Is the manuscript technically sound, and do the data support the conclusions?

Reviewer #1: (No Response)

Reviewer #3: Yes

3. Has the statistical analysis been performed appropriately and rigorously? 

Reviewer #1: (No Response)

Reviewer #3: Yes

4. Have the authors made all data underlying the findings in their manuscript fully available?

Reviewer #1: (No Response)

Reviewer #3: Yes

5. Is the manuscript presented in an intelligible fashion and written in standard English?

Reviewer #1: (No Response)

Reviewer #3: Yes

6. Review Comments to the Author

Reviewer #1: Why don't the pictures have numbers?

The article's text has repetitions in both sentences and content.

Reviewer #3: I think the authors addressed the comments raised by previous reviewers and it is now in good shape to be published

7. PLOS authors have the option to publish the peer review history of their article (what does this mean? ). If published, this will include your full peer review and any attached files.

**Do you want your identity to be public for this peer review?** For information about this choice, including consent withdrawal, please see our Privacy Policy .

Reviewer #1: **Yes: ** Somayeh Mehrabadi

Reviewer #3: No

---

## [Author Response · Author response to Decision Letter 3]

9 May 2025

Thank you for your valuable review comments and suggestions! We have deleted the repeated statements in the text. At the same time, we have numbered and remarked the system uploaded. In order to facilitate your review, we have re-inserted them below. Please review, Thank you again!

---

## [Editor Report · Decision Letter 3]

Identification of Fresh Leaves of Anji White Tea : S-YOLOv10-ASI Algorithm Fusing Asymptotic Feature Pyra-mid Network

PONE-D-24-60012R3

Dear Dr. Wang,

We’re pleased to inform you that your manuscript has been judged scientifically suitable for publication and will be formally accepted for publication once it meets all outstanding technical requirements.

Kind regards,

Yile Chen, Ph.D. in Architecture

Academic Editor

PLOS ONE
---

## [Editor Report · Acceptance letter]

PONE-D-24-60012R3

PLOS ONE

Dear Dr. Wang,

I'm pleased to inform you that your manuscript has been deemed suitable for publication in PLOS ONE. Congratulations! Your manuscript is now being handed over to our production team.

Kind regards,

on behalf of

Dr. Yile Chen

Academic Editor

PLOS ONE